# Revealing the Critical Role of Global Electron Density Transfer in the Reaction Rate of Polar Organic Reactions within Molecular Electron Density Theory

**DOI:** 10.3390/molecules29081870

**Published:** 2024-04-19

**Authors:** Luis R. Domingo, Mar Ríos-Gutiérrez

**Affiliations:** Department of Organic Chemistry, University of Valencia, Dr. Moliner 50, 46100 Burjassot, Valencia, Spain

**Keywords:** global electron density transfer, molecular electron density theory, polar reactions, interacting quantum atoms, interacting quantum fragments, theoretical chemistry

## Abstract

The critical role of global electron density transfer (GEDT) in increasing the reaction rate of polar organic reactions has been studied within the framework of Molecular Electron Density Theory (MEDT). To this end, the series of the polar Diels–Alder (P-DA) reactions of cyclopentadiene with cyanoethylene derivatives, for which experimental kinetic data are available, have been chosen. A complete linear correlation between the computed activation Gibbs free energies and the GEDT taking place at the polar transition state structures (TSs) is found; the higher the GEDT at the TS, the lower the activation Gibbs free energy. An interacting quantum atoms energy partitioning analysis allows for establishing a complete linear correlation between the electronic stabilization of the electrophilic ethylene frameworks and the GEDT taking place at the polar TSs. This finding supports Parr’s proposal for the definition of the electrophilicity ω index. The present MEDT study establishes the critical role of the GEDT in the acceleration of polar reactions, since the electronic stabilization of the electrophilic framework with the electron density gain is greater than the destabilization of the nucleophilic one, making a net favorable electronic contribution to the decrease in the activation energy.

## 1. Introduction

One of the most useful classifications of organic reactions is into non-polar and polar reactions. While non-polar reactions are experienced mainly by hydrocarbons, most polar reactions are characteristic of organic molecules presenting carbon-heteroatom functional groups. Unsaturated hydrocarbon compounds also experience polar reactions when they are adequately substituted by electron-withdrawing or electron-releasing groups. The polar character determines the rate of an organic reaction; while non-polar reactions have high activation energies, in general the more polar, the faster the reaction.

The idea of polar reactions in organic chemistry was introduced at the beginning of the last century. The introduction of the terms ‘nucleophile’ and ‘electrophile’ as chemists use them today is officially attributed to C. K. Ingold [1], who replaced the terms ‘anionoid’ and ‘cationoid’ proposed earlier by A. J. Lapworth in 1925 [2]. The nucleophilicity and electrophilicity concepts were further extended to reactions, and thus there are now so-known electrophilic and nucleophilic attacks, additions, substitutions, etc. [3].

From a theoretical point of view, the implementation in 1999 of the Parr’s electrophilicity ω index [4] within conceptual DFT [5,6] was a breakthrough in the theoretical understanding of polar reactions. Thus, in 2002, in an attempt to classify the Diels–Alder (DA) reactions within the polar organic reactions, the first scale of the Parr’s electrophilicity ω index of organic molecules was established, allowing to explain the substituent effects on the diene and the ethylene in the reaction rate [7]. The polar reactivity model of DA reactions was extended to [2+2], [3+2], [5+2], and so on, cycloaddition reactions. After the study of many organic reactions, the empirical nucleophilicity *N* index was proposed in 2008 [8] which, together with the electrophilicity ω index, have become powerful tools to study the polar reactivity in organic chemistry [9].

In 1964, Sauer et al. [10] experimentally reported the reactivity of Cp **1** with the cyanoethylene series **2** in DA reactions (see Figure 1). The reported experimental reaction rate constants have been used in organic chemistry textbooks to show how the electron-withdrawing substitution on ethylene increases the reaction rates of DA reactions (see Table 1) [11].

The Sauer’s DA reactions of the cyanoethylene series have been theoretically studied [11,12]. A very good linear correlation between the global electron density transfer [13] (GEDT) taking place at the transition state structures (TSs) [11] and the logarithm of the experimental reaction rate constants [10] was established for the first time in 2009 [14], indicating that the GEDT could be one of the key factors in the activation energy (see Figure 1).

After many theoretical studies of experimental DA reactions, a very good linear correlation between the activation energies of DA reactions and the GEDT taking place at the TSs was recognized in 2009, thus establishing the general mechanism of polar Diels–Alder (P-DA) reactions [14], which are controlled by the nucleophilic and electrophilic interactions taking place at the TSs. In polar reactions, the increase in GEDT at the TSs causes a decrease in the activation energies, increasing the reaction rates [11,14]. This behavior has been found in many polar organic reactions such as [2+2] and [3+2] cycloaddition reactions, Lewis acid-catalyzed DA reactions, higher-order cycloaddition reactions, Michael reactions, Wittig reactions, and electrophilic aromatic substitution reactions.

In 2017, a comparative electron localization function [15] (ELF) and Quantum Theory of Atoms in Molecules [16,17] (QTAIM) topological analysis of the TSs of the DA reactions of Cp **1** with ethylene **4** and with tetracyanoethylene **4CN** [18] showed that in the polar reaction involving **4CN**, the GEDT favors the bonding changes on the reagents demanded for the formation of the new C–C single bonds [13], allowing the TS to occur earlier [18]. However, the effects of the GEDT in the reduction in the activation energies could not be established.

In 2016, Domingo proposed the Molecular Electron Density Theory [19] (MEDT) for the study of chemical organic reactivity. MEDT establishes that changes in electron density along a reaction, and not molecular orbital interactions, are responsible for chemical reactivity. Accordingly, MEDT rejects any model based on molecular orbital analyses, such as the Frontier Molecular Orbital theory [20] and Morokuma-based energy decomposition schemes [21,22], all of which were developed in the last century.

In 2005, Blanco et al. developed a parameter-free and reference-free energy decomposition scheme based on QTAIM, called interacting quantum atoms [23] (IQA). IQA divides the total energy of a system into intra- and inter-atomic contributions, each of a different nature, which are related to chemical concepts of bonding. IQA analysis has very recently been used in MEDT studies of [3+2] cycloaddition reactions [24] and P-DA reactions [25].

Due to the relevance of the GEDT taking place at the TSs of polar reactions in organic chemistry [11], an MEDT/IQA study of the P-DA reactions of Cp **1** with the Sauer’s cyanoethylene series **2**, as models of polar organic reactions, is herein performed in order to definitively establish the critical role of GEDT in the acceleration in polar processes both qualitatively and quantitatively. To this end, an IQA energy partitioning analysis at the TSs and reagents is carried out. The conclusions of this study can be extrapolated to any organic polar reaction.

## 2. Results and Discussion

This section has been divided in the following subsections: (i) first, the DFT-based reactivity indices of the reagents are analyzed; (ii) next, kinetic, thermodynamic, and geometrical parameters associated with the P-DA reactions of Cp **1** with the cyanoethylene series **2** are discussed, along with some correlations involving electrophilicities and GEDT values; and finally, (iii) a density-based IQA energy partitioning analysis is performed in order to establish, both quantitatively and qualitatively, the role of GEDT in increasing the reaction rates with the cyano substitution on the ethylene. Except for kinetic and thermodynamic data, which are analyzed in dioxane for comparison with the experiment, all other analyses are discussed in vacuo for coherence with the IQA analysis, which does not yet support the implicit effect of solvent in atomic integrations (see the Appendix A).

### 2.1. Analysis of the Reactivity Indices at the Ground State of the Reagents

The reactivity indices [5,6] for Cp **1** and the cyanoethylene series **2** computed at the ground state are gathered in Table 2. Computational details are given in Appendix A. The B3LYP/6-31G(d) computational method was used because the original reactivity scales were established at that level [5,6]. Analysis of the electronic chemical potentials [26] μ of the regents allows for establishing of the polar character of an organic reaction, as well as the unambiguous establishment of the direction of the flux of the electron density, which enables the classification of polar reactions [27,28]. The electronic chemical potential μ of Cp **1**, −3.01 eV, is above those of the cyanoethylene series **2**, between −4.70 (**1CN**) and −7.04 (**4CN**) eV, indicating that in these polar reactions the electron density will flux from Cp **1** towards these cyanoethylenes, being classified as P-DA reactions of forward electron density flux [27,28].

Cp **1** presents an electrophilicity ω index [4] of 0.83 eV and a nucleophilicity *N* index [8] of 3.37 eV, being classified as a moderate electrophile and a strong nucleophile. Consequently, Cp **1** will participate in P-DA reactions only as a good nucleophile. On the other hand, ethylene **4** presents an electrophilicity ω index of 0.73 eV and a nucleophilicity *N* index of 1.87 eV, being classified as a marginal electrophile and a marginal nucleophile. Consequently, ethylene **4** will never participate in a P-DA reaction. The non-polar Diels–Alder (N-DA) reaction of Cp **1** with ethylene **4** is classified within MEDT as a null electron density flux reaction [28].

The electrophilicity ω index of cyanoethylene series **2** ranges from 1.74 eV (**1CN**) to 5.95 eV (**4CN**), while the nucleophilicity *N* index ranges from 1.25 eV (**1CN**) to 0.00 eV (**4CN**). Note that the nucleophilicity *N* index for **4CN** is exactly 0.00 eV because this molecule was chosen as the reference for the empirical nucleophilicity *N* scale [6,9]. Thus, while **1CN** is located on the borderline between moderate electrophiles, the other cyanoethylenes are clearly classified as strong electrophiles; note that the tri- and tetracyanoethylenes **3CN** and **4CN**, with ω > 4.0 eV, are classified as superelectrophiles, a behavior that accounts for their high reactivity in polar processes [9] (see Table 1). On the other hand, all cyanoethylenes are classified as marginal nucleophiles. Consequently, this cyanoethylene series will participate towards Cp **1** in P-DA reactions of forward electron density flux [27,28]. Both electrophilic and nucleophilic properties of this series vary with the number of cyano groups on the ethylene.

Along a polar cycloaddition reaction involving non-symmetric species such as **1CN**, **2CN** or **3CN**, the most favorable reaction path involves the two-center interaction with the most electrophilic center of these cyanoethylenes [11]. In this sense, the analysis of the electrophilic $P_{k}^{+}$ Parr functions [29] of the cyanoethylene series is a valuable tool to characterize the most electrophilic center of these molecules (see Figure 2 and Table 3).

Analysis of the electrophilic
$P_{k}^{+}$ Parr functions at the C1 and C2 carbons of the cyanoethylene series **2** indicates that the two ethylene carbons gather more than 60% of the total amount of spin density in these molecules; i.e., the two carbons will accumulate more than 60% of the electron density transferred to these ethylene derivatives via the GEDT in these P-DA reactions. As expected, the symmetrically substituted ethylenes **2cCN**, **2tCN,** and **4CN** present identical electrophilic $P_{k}^{+}$ Parr functions at the two carbons, while the non-symmetrically substituted **1CN**, **2CN**, and **3CN** present a non-symmetrical electrophilic activation; in the three cases, the less-substituted carbon present the higher electrophilic $P_{k}^{+}$ Parr function (see Table 3).

Analysis of the local electrophilicity ω_k_ indices [30] at the C1 and C2 carbons of these cyanoethylenes allows for obtaining some appealing conclusions (see Table 3): (i) while the symmetrically substituted ethylenes **2cCN**, **2tCN** and **4CN** present identical electrophilic activation at the two ethylene carbons, predicting synchronous TSs, the non-symmetrically substituted **1CN**, **2CN** and **3CN** present different electrophilic activation, predicting asynchronous TSs [11]; (ii) at the non-symmetrically substituted **1CN**, **2CN** and **3CN** cyanoethylenes, the less-substituted C1 carbon presents the higher electrophilic activation, indicating that this carbon will be the preferred center to participate in the two-center interaction with the C6 carbon of the nucleophilic Cp **1** [11]; and finally, (iii) the local electrophilicity at the C1 carbon of **2CN**, ω_k_ = 2.28 eV, is higher than that at the C1 carbon of **3CN**, ω_k_ = 2.03 eV, despite the more electrophilic character of **3CN** than **2CN**. Note that the two C1 and C2 carbons of the symmetrically substituted **2cCN** and **2tCN** are electrophilically activated by ca. ω_k_ = 1.0 eV, for each one.

### 2.2. Study of the P-DA Reactions of Cp **1** with the Cyanoethylene Series **2**

For the non-symmetrically substituted cyanoethylenes **1CN**, **2CN,** and **3CN**, two stereoisomeric reaction paths are feasible; only the *endo* approach mode was studied herein (see Figure 1). A detailed analysis of the potential energy surfaces is found in [11]. The M06-2X/6-311G(d,p) relative Gibbs free energies of TSs and CAs in dioxane are given in Table 4, while complete thermodynamic data are given in Appendix A.

The activation Gibbs free energies range from 25.9 (**TS-1CN**) to 12.7 (**TS-4CN**) kcal mol^−1^. Note that the N-DA reaction of Cp **1** with ethylene **4** displays a very high activation Gibbs free energy of 29.8 kcal mol^−1^ (see Table 4). These DA reactions are exergonic in the narrow range between –12.7 (**CA-1CN**) and –14.9 (**CA-4CN**) kcal mol^−1^.

A representation of the activation Gibbs free energies versus the number of cyano groups on the ethylene shows a very good linear correlation with a coefficient of determination R^2^ = 0.94 (see Appendix A). This graph shows that the presence of the cyano group on the ethylene is additive and has a marked effect on the kinetics of the reactions, in clear agreement with the experimental outcomes observed by Sauer et al. (see Table 1).

Using the Eyring–Polanyi equation [31], the relative reaction rate constants k_r_ of the P-DA reactions between Cp **1** and the cyanoethylene series **2**, with respect to that with ethylene **4**, were computed (see Table 4). The relative reaction rate constants k_r_ range from 8.09 × 10^2^ (**1CN**) to 5.62 × 10^12^ (**4CN**). Thus, the P-DA reaction involving the superelectrophilic tetracyanoethylene **4CN** is 10^12^ faster than the N-DA reaction of Cp **1** with ethylene **4**. A representation of the logarithm of the experimental relative reaction rate constants log(k_rexp_), with respect to the N-DA reactions of Cp **1** with ethylene **4**, versus the logarithm of the computed relative reaction rate constants log(k_rcomp_) shows a complete linear correlation with an R^2^ = 1.00 (see Appendix A).

The main geometrical parameters at the in vacuo TSs, i.e., the distances between the two pairs of C1–C6 and C2–C3 interacting centers, together with the geometrical asynchronicity, **Δ**l, and the average of the two C–C distances, lm, are given in Table 5. Geometrical data in dioxane are gathered in Appendix A; they show no significant changes compared to the in vacuo parameters. The geometries of two representative TSs are given in Figure 3, while the geometries of all TSs are given in Appendix A. Some appealing conclusions can be obtained from the geometrical data given in Table 5: (i) from a geometrical point of view, the TSs can be classified as synchronous and asynchronous TSs, depending on the evolution of the new C–C single bond formation; (ii) while the synchronous TSs, **Δ**l = 0.0 Å, come from the symmetrically substituted ethylenes, asynchronous TSs, **Δ**l > 0.2 Å, come from the non-symmetrically substituted ethylenes; (iii) interestingly, the average of the two C1–C6 and C2–C3 distances at all TS, including **TS-Et**, is 2.24 Å (see lm in Table 5). This behavior indicates that all TSs have a comparable advanced/early character. Considering that the C–C single bond formation takes place in the short range of 2.0–1.9 Å [13], these geometrical parameters indicate that formation of the first C–C single bond has not yet started in any of the TSs (see later). This behavior is consistent with Woodward’s 1942 proposal that electron transfer occurs before the formation of the new C-C single bonds [32].

As Figure 4 shows, a complete linear correlation between the geometrical asynchronicity, Δl, of the TSs and the difference of the local electrophilicity ω_k_ indices of the C1 and C2 carbons of the cyanoethylenes, Δω_k_, is established for the first time, with R^2^ = 1.00. Thus, the symmetrically substituted cyanoethylenes **2cCN**, **2tCN** and **4CN** with Δω_k_ = 0.00 eV yield synchronous TSs, while non-symmetrically substituted cyanoethylenes **1CN**, **2CN** and **3CN** with Δω_k_ > 0.76 eV yield asynchronous TSs. This excellent relationship indicates that the different electrophilic activation of the two C1 and C2 carbons of the electrophilic ethylenes caused by the electron-withdrawing substitution controls the asynchronicity of the C–C single bond formation in these P-DA reactions.

Analysis of GEDT at the TSs permits to quantify the polar character of these DA reactions [14]. The GEDT values computed at the seven TSs in vacuo are given in Table 5. GEDT values in dioxane are displayed in Appendix A and show no significant changes compared to the in vacuo values. As expected, the GEDT value at **TS-Et** is negligible, 0.03 e, as a consequence of the marginal electrophilic character of ethylene **4** (see Table 2). Consequently, the corresponding DA reaction has a non-polar character, being classified as null electron density flux [28]. The presence of a cyano group in **1CN** notably increases the GEDT at **TS-1CN** to 0.14 e. The addition of cyano groups at the ethylene moiety markedly increases the GEDT at the corresponding TSs, reaching a maximum value at **TS-4CN** with a GEDT = 0.42 e. These P-DA reactions are classified as FEDF [27,28], in agreement with the previous reactivity indices prediction (see Section 2.1). The presence of at least two cyano groups makes the corresponding DA reaction very polar (see Table 5). It is worth mentioning that GEDT values obtained from an NPA analysis do not significantly vary with the charge partitioning method because of its formal definition (see Appendix A). For instance, in vacuo GEDT values computed with Bader charges show no significant variation (see in Appendix A and the linear regression in Appendix A).

A representation of the activation Gibbs free energies of these P-DA reactions versus the computed GEDT values at the corresponding TSs also shows a very good linear correlation with an R^2^ = 0.98 (see Figure 5). This linear correlation, that has been found in numerous organic reactions, shows the significant role of the polar character of the reactions, measured by the computed GEDT values, in reaction rates.

It is worth noting that although geminal **2CN** is less electrophilic than vicinal **2cCN** and **2tCN**, the reaction of **2CN** is more polar and has a higher reaction rate (see Figure 5). This finding points out, once again, the relevant role of GEDT in reaction rates, and indicates that asynchronous processes are generally preferred over synchronous ones due to a more favorable two-center interaction (see later).

Finally, ELF and QTAIM topological analyses of the electronic structures of the reagents and TSs were performed. The corresponding analyses are given in Appendix A. ELF analysis of the reagents indicates that the cyano substitution on the ethylene does not cause any remarkable changes in the C–C double bond region at the ground state of these substituted ethylenes. On the other hand, ELF analysis of the TSs indicates that while the low polar **TS-1CN** and **TS-2cCN** and **TS-2tCN** show a great similitude to the non-polar **TS-Et**, the highly polar **TS-2CN**, **TS-3CN** and **TS-4CN** show the presence of the *pseudoradical* centers demanded for the subsequent C–C single bond formation [13]. Both ELF and QTAIM analyses of the electron density at the TSs indicate that formation of the C–C single bonds has not started yet in any of them, thus rejecting the concept of concerted TSs.

### 2.3. IQA Analysis of the TSs of the P-DA Reactions of the Cyanoethylene Series **2**

In order to determine the role of the GEDT caused by the cyano substitution on the ethylene in the experimental acceleration observed by Sauer et al. (see Table 1), a topological IQA [23] energy partitioning was carried out at the seven TSs in vacuo. For this purpose, an interacting quantum fragments approach [33] was adopted, considering relative IQA energies at both interacting frameworks as defined in Appendix A. The relative total, intra- and inter-atomic IQA energies of each TS fragment are given in Table 6, while the total values are given in Appendix A.

Table 6 shows that the stabilization of the ethylene framework with the number of cyano groups, ΔE_tot_(nCN) < 0, is stronger than the Cp destabilization, ΔE_tot_(Cp) > 0, justifying the decrease in the activation energies along this cyanoethylene series. In addition, while the increase in the cyano substitution in the ethylene generally increases E_intra_(X) in the two interacting frameworks and V_inter_(Cp), a huge decrease in the inter-atomic V_inter_(nCN) energies, by between −16.0 (**1CN**) and −102.0 (**4CN**) kcal mol^−1^, is observed (see the differences between V_inter_(nCN) and V_inter_(Et)).

Figure 6 shows a graphical representation of the interacting quantum fragment E_tot_(X) energy of the Cp and Et frameworks at the seven TSs and the E_tot_(Cp+Et), which corresponds to the activation energies of these DA reactions. As can be observed, both E_tot_(X) are positive and unfavorable in the N-DA reaction with ethylene **4**. However, along the cyanoethylene series **2**, the interacting quantum fragment energies associated with the Cp framework increase while those of the ethylene derivatives become more negative, i.e., more stabilizing. This stabilization reaches such an extent that, in **TS-4CN**, the stabilization of the ethylene framework overcomes the destabilization of the Cp one, and the corresponding relative energy of **TS-4CN** becomes negative. These behaviors are a consequence of the GEDT that takes place at the TSs (see Figure 6), that while it destabilizes the nucleophile Cp for making it lose electron density, it stabilizes the electrophile nCN as it gains electron density.

Consequently, the strong stabilization of the cyanoethylene framework at the TSs with the cyano substitution, by between 14.7 (**1CN**) and 62.5 (**4CN**) kcal mol^−1^ with respect to the N-DA reaction of Cp **1** with ethylene **4**, accounts for the decrease in the activation energies associated with the P-DA reactions between Cp **1** and the cyanoethylene series **2**.

A representation of the logarithm of the experimental reaction rate constant k versus the stabilization of the ethylene frameworks at the TSs shows an excellent linear correlation with an R^2^ = 0.95 (see Appendix A). This figure shows the close relationship between the experimentally observed acceleration in this series of P-DA reactions and the decrease in activation energy resulting from the electronic stabilization of the ethylene framework. Furthermore, a representation of E_tot_(Et) versus the GEDT computed at the TSs shows an excellent linear correlation with an R^2^ = 0.99 (see Figure 7).

These linear correlations allow establishing, for the first time, that the electronic stabilization of the ethylene framework, resulting from the GEDT process taking place in polar reactions, is responsible for the increase in the reaction rate observed in these polar reactions [14]. Note that the N-DA reaction of ethylene **4**, which presents a GEDT = 0.03 e, fits in the top left corner in the linear regression in Figure 7.

As the energy factor that changes the most with the cyano substitution is V_inter_(Et), in order to gain a more detailed insight into the stabilization of the ethylene derivatives, the inter-atomic interactions between the Cp and ethylene frameworks, V_inter_(Cp,Et), were considered separately from the interactions that take place within each of them, V’_inter_(X). The corresponding energies, together with the standard deviations with respect to the N-DA reaction of ethylene **4** as the reference, are given in Table 7.

The standard deviation values indicate that the most drastic changes with the cyano substitution take place in decreasing the inter-atomic interactions occurring inside the ethylene fragment (see the standard deviation of V′_inter_(Et) in Table 7). This effect overcomes the changes in the inter-atomic interactions between the fragments, V_inter_(Cp,Et), which also become more stabilizing as the polar character of the reaction increases.

These findings confirm that the stabilization of the electrophilic reagent in P-DA reactions is the most relevant consequence of the GEDT, thus being responsible for the increase in reaction rates with the increase in the polar character [13].

Finally, in 1999, Parr proposed the electrophilicity ω index as a measure of the electronic stabilization of a species when it acquires a certai n amount of electron density from the environment [4]. Thus, when the sum of the relative intra-atomic E_intra_(Et) and inter-atomic V′_inter_(Et) IQA energies, i.e., E′_tot_(Et), are represented versus the corresponding Parr’s electrophilicity ω indices (see Table 2), a very good linear correlation is obtained; R^2^ = 0.97 (see Figure 8). The E′_tot_(Et) in the N-DA reaction between Cp **1** and ethylene **4** is very unfavorable, 84.6 kcal mol^−1^. The inclusion of the cyano groups stabilizes the ethylene framework at the TSs by between 11.3 (**1CN**) and 45.7 (**4CN**) kcal mol^−1^ as a consequence of the GEDT taking place at the polar TSs (see Figure 8). Consequently, this graph supports Parr’s proposal [4,34]; the higher the electrophilicity ω index, the higher the ethylene stabilization at the polar TSs. Given that the ethylene stabilization via the GEDT is the main factor responsible for the decrease in activation energies, as shown above, this linear correlation also validates Parr’s electrophilicity ω index as a solid predictor of reactivity in polar cycloaddition reactions.

## 3. Computational Methods

The M06-2X functional [35], together with the standard 6-311G(d,p) basis set [36], which includes d-type polarization for second-row elements and p-type polarization functions for hydrogens, was used throughout this MEDT study. In general, a careful study of DA reactions calls for the adoption of higher-level wave function models, placing more emphasis on the electron correlation effects than single-determinantal wave function methods. However, the polar character of the DA reactions studied in the present manuscript enables the single-determinant level. The TSs were characterized by the presence of only one imaginary frequency.

The solvent effects of dioxane were taken into account in the thermodynamic calculations by full optimization of the in vacuo structures at the same computational level using the polarizable continuum model [37,38] in the framework of the self-consistent reaction field [39,40,41]. The values of M06-2X/6-311G(d,p) enthalpies, entropies, and Gibbs free energies in dioxane were calculated with standard statistical thermodynamics [36] at 293.15 °C and 1 atm by polarizable continuum model frequency calculations at the solvent-optimized structures. Except for the thermodynamic data, the rest of the results are obtained in in vacuo for an overall coherence in interpreting the results obtained using different quantum chemical tools.

DFT reactivity indices were calculated using the equations in [6]. The GEDT [13] values were computed using the equation GEDT(f) = Σq_f_, where q are the natural charges [42,43] of the atoms belonging to one of the two frameworks (f) at the TS geometries.

The Gaussian 16 suite of programs was used to perform the calculations [44]. Molecular geometries, ELF basin attractors, and the 3D representations of the Mulliken atomic spin densities were visualized by using the GaussView program [45]. ELF analyses [15] of the M06-2X/6-311G(d,p) monodeterminantal wavefunctions were performed by using the TopMod [46] package with a cubical grid of step size of 0.1 Bohr. The Bader’s QTAIM [16,17] analyses were conducted using Multiwfn 3.7 software packages [47] The IQA [35] analysis was performed with the AIMAll package [48] using the corresponding M06-2X/6-311G(d,p) monodeterminantal wavefunctions.

## 4. Conclusions

An IQA energy partitioning analysis of the TSs associated with the P-DA reactions of Cp **1** with the cyanoethylene series **2** permits to establish the decisive role of GEDT in the acceleration found in polar reactions. The topological IQA energy partitioning allows establishing the complete linear correlation between the total IQA atomic energies of the electrophilic ethylene framework and the GEDT taking place at the TSs of these P-DA reactions. This finding establishes that the increase in GEDT at the TSs enhances the reaction rates of P-DA reactions through an electronic stabilization of the electrophilic framework.

The present MEDT study also shows that the asynchronicity in P-DA reactions depends on the non-symmetric electrophilic activation of the two interacting carbons of the ethylene caused by the electron-withdrawing substitution. A very good linear correlation between the geometric asynchronicity, Δl, at the TSs and the difference in the local electrophilicity ω_k_ indices of the C1 and C2 carbons of the cyanoethylenes, Δω_k_, is found.

## Data Availability

The data that support the findings of this study are available from the corresponding author upon reasonable request.

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
