# Peer review of "Revealing the Critical Role of Global Electron Density Transfer in the Reaction Rate of Polar Organic Reactions within Molecular Electron Density Theory"

_molecules, 2024, doi:10.3390/molecules29081870_

Round 1

Reviewer 1 Report

Comments and Suggestions for Authors

The manuscript deals with a detailed analysis of using Global Electron Density Transfer (GEDT) as a descriptor in polar organic reactions: increase in GEDT is associated with increase of reaction rates. The manuscript is of interest to a broad audience (both theoreticians and experimentalists). It is well written, and the conclusions are supported by the results. I suggest the manuscript to be accepted in the current form.

Author Response

We thank the comments of referee 1.

Reviewer 2 Report

Comments and Suggestions for Authors

The ideas presented her are interesting and can be published but the following details that must be corrected before publication:

- The authors use many abbreviations that make reading difficult. Authors should reduce the use of abbreviations to a minimum or better not use any.

- The introduction is very long. There is a lot of historical information that is not necessary to have the context of the research. It should go from general to specific information, ending with what is important and missing that the authors will present in this article.

- The conclusion should be the “take home message,” not a summary of the results. They should be one or two paragraphs with the most important ideas that emerged from the results.

- It is imperative to include the questions that the authors want to answer with this research. They are probably included in the current version, but in a very confusing way.

Comments on the Quality of English Language

Check the grammar. For example, the first paragraph of the introduction has the word "reactions" many times.

Author Response

The ideas presented her are interesting and can be published but the following details that must be corrected before publication: 

- The authors use many abbreviations that make reading difficult. Authors should reduce the use of abbreviations to a minimum or better not use any.

R. The original manuscript contained twenty abbreviations commonly used in MEDT studies.

As suggested by the reviewer, the abbreviations that were not repeated more than ten times were removed. Thus, a total of twelve abbreviations have been eliminated in this revised version. 

- The introduction is very long. There is a lot of historical information that is not necessary to have the context of the research. It should go from general to specific information, ending with what is important and missing that the authors will present in this article.

R. In agreement with the referee’s suggestion, some paragraphs and schemes of the introduction part have been removed in this revised version.

- The conclusion should be the “take home message,” not a summary of the results. They should be one or two paragraphs with the most important ideas that emerged from the results.

R. As suggested by the referee, the Conclusion section has been reduced to two paragraphs stating the main conclusions of the manuscript.

- It is imperative to include the questions that the authors want to answer with this research. They are probably included in the current version, but in a very confusing way.

R. The main question that has been resolved in this manuscript is understanding the decisive role of the GEDT in the reduction of the activation energies in polar organic reactions.

Along the introduction, the relevance of the GEDT in the activation energy observed in organic reactions is pointed out.

In the last paragraph of the Introduction part, the main question that we want to answer in this MEDT study was included in the original manuscript.

Reviewer 3 Report

Comments and Suggestions for Authors

This manuscript by Luis R. Domingo and Mar Ríos-Gutiérrez is aimed at disclosing the critical role of global electron density transfer (GEDT) in increasing the reaction rate of the polar Diels-Alder (DA) reactions, using several tools based on electron densities (electron and electron pair densities, the latter required by IQA) and density matrices (needed for ELF).

The reported work has several merits; it is carefully performed, and the results obtained are cleverly discerned/distilled and clearly reported.  The obtained results are also of relevant importance.

This manuscript is surely publishable in Molecules, almost in the present form, after the authors consider these few minor points/observations:

 a)     In general, a careful study of DA reactions calls for the adoption of higher level wavefunctiuon models, taking better into account the electron correlation effects, than can the “simple” single-determinantal wf methods used in the present study. However, the polar character of the DA reactions studied in the present manuscript, enables the authors to restrain themselves to the single-determinant level.  This should probably be said explicitly by the authors at a convenient point in their manuscript.

b)     The term “gas-phase” is everywhere used in the manuscript to refer to what are instead “in-vacuo” calculations. Although it is common practice to use the “gas-phase” diction also in such a case, I think that the correct diction should be used, also for a proper distinction relative to the”in solvent” calculations case.

c)     Figure 2: which is the true reason to plot atomic Mulliken spin densities? Should not be more reasonable to directly plot the electron spin density itself without introducing useless biases due to Mulliken partitioning? (even though I do not expect significant changes)

d)     Figure 5, caption: I guess that “R2 = 0.89” should rather be “R2 = 0.98”.

e)     Page 10, line 331: “….as defined in the Supplementary Material”

f)      Page 14, line 466: “…GEDT   taking place at…” rather than “…GEDT taking places ta…”

Comments on the Quality of English Language

English is fine and clear

Author Response

This manuscript is surely publishable in Molecules, almost in the present form, after the authors consider these few minor points/observations:

a) In general, a careful study of DA reactions calls for the adoption of higher level wavefunctiuon models, taking better into account the electron correlation effects, than can the “simple” single-determinantal wf methods used in the present study. However, the polar character of the DA reactions studied in the present manuscript, enables the authors to restrain themselves to the single-determinant level.  This should probably be said explicitly by the authors at a convenient point in their manuscript.

R. In agreement with the referee’s suggestion, the suitability of single-determinant wave functions for the studied polar DA reactions is explicitly mentioned in the Computational Methods section.

b) The term “gas-phase” is everywhere used in the manuscript to refer to what are instead “in-vacuo” calculations. Although it is common practice to use the “gas-phase” diction also in such a case, I think that the correct diction should be used, also for a proper distinction relative to the”in solvent” calculations case.

R. As suggested by the referee, "gas phase" has been replaced by "in-vacuo" in the manuscript and supplementary material.

c)  Figure 2: which is the true reason to plot atomic Mulliken spin densities? Should not be more reasonable to directly plot the electron spin density itself without introducing useless biases due to Mulliken partitioning? (even though I do not expect significant changes)

R. This figure shows the 3D representations given by the Gauss View program of the Mulliken atomic spin densities which are used to compute the Parr functions (see RSC Adv. 2013, 3, 1486). This behavior has been mentioned in the Computational Methods section in this revised version.

d)     Figure 5, caption: I guess that “R= 0.89” should rather be “R= 0.98”.

R. Thanks. This error in the caption of Figure 5 has been corrected.

  1. e)     Page 10, line 331: “….as defined in theSupplementary Material”

  1. Thanks. This error in the caption of Figure 5 has been corrected.

  1. f) Page 14, line 466: “…GEDT   taking place at…” rather than “…GEDT taking places ta…”
  2. Thanks. This error in the caption of Figure 5 has been corrected.